# An All Optical 2 × 1 Multiplexer Using a Metal-Insulator-Metal based Plasmonic Waveguide for Processing at a Rapid Pace

Ipshitha Charles [1], Sandip Swarnakar [1,*], Geetha Rani Nalubolu [2], Venkatrao Palacharla [3] and Santosh Kumar [4,*]

1 Photonics Laboratory, Department of Electronics and Communication Engineering, G. Pullaiah College of Engineering and Technology, Nandikotkur Road, Kurnool 518002, India
2 Department of Electronics and Communication Engineering, Ravindra College of Engineering for Women, Nandikotkur Road, Kurnool 518452, India
3 Godavari Institute of Engineering & Technology, Department of Electronics & Communication Engineering, Rajahmundry 533296, India
4 Shandong Key Laboratory of Optical Communication Science and Technology, School of Physics Science and Information Technology, Liaocheng University, Liaocheng 252059, China
* Correspondence: drsandipece@gpcet.ac.in (S.S.); santosh@lcu.edu.cn (S.K.)

**Abstract:** This study proposes, designs, and simulates a unique plasmonic Y-shaped MIM waveguide based 2 × 1 multiplexer (MUX) structure utilising opti-FDTD software. Two plasmonic Y-shaped waveguides are positioned facing one another inside a minimum wafer size of 6 μm × 3.5 μm in the 2 × 1 MUX configurations that is being described. The design parameters are adjusted until the plasmonic multiplexer performs as required under optimal conditions. Extinction ratio and insertion loss are two performance metrics that are calculated for performance analysis of the design, which indicate the potential to be applied in plasmonic integrated circuits.

**Keywords:** multiplexer; MIM waveguide; Y-shaped Waveguide; finite-difference time-domain (FDTD); plasmonic waveguide





## 1. Introduction

The operating speed issue with its counterpart, the electrical circuits, has been widely addressed by all-optical systems [1,2]. As the bandwidth and bit rate of electronic-based processing and computing systems approach their limits, future optical communications and networks will require all-optical data processing. Researchers used a variety of techniques to create all-optical devices, including Kerr materials, Mach-Zehnder interferometers, the self-collimation approach, optical rings resonators, photonic crystals, and plasmonic waveguide [3–9]. This raised the need for photonics since it severely controls light and only requires a very minimal input power to switch on [10–16]. In addition to being faster and compact, these optical circuits are equivalent electrical circuits in size. The diffraction limit of photonic circuits has enabled a new field, plasmonics [17], which integrates photonics with electronics at the nanoscale and has attracted a lot of attention due to its reduced diffraction limit and higher frequencies, which allow for faster data transmission. The generation, detection, and manipulation of optical signals at the material interface are the main goals of plasmonics [18]. The primary drawbacks of plasmonic circuits include their limited propagation length, high heat emission, and difficulty in changing a signal's direction within the circuit [19]. In order to reduce these losses, plasmonic waveguides are used in these circuits to optimise both the length of surface plasmon propagation and confinement. Additionally, the fact that they can work in the visible to far-infrared range while using less power and a faster processing speed is drawing a lot of interest [20–22].

Numerous plasmonic waveguide types have been studied by researchers, including Metal-Insulator-Metal (MIM) [23–25], Insulator Metal Insulator (IMI) [26–29], and hybrid waveguides [30–32]. Plasmonic MIM waveguides, which have a dielectric core and

two metallic cladding layers and exhibit more confinement than insulating waveguides, are presented as a potential choice for nanoscale optical circuits since they outperformed insulating waveguides in this regard. Multiple experiments on a variety of MIM devices, such as logic gates, switches, couplers, splitters, and de-multiplexers, validated SPP modes' reliable localization over a broad range of wavelengths with long propagation distance and ease of manufacturing [33–36]. MIM waveguides are now often utilised in sensors, logic gates, filters, lenses, and switches. Two-dimensional (2-D) MIM waveguides were chosen for the proposed logic device due to their straightforward design, ability to confine light at the microscopic scale, reduced crosstalk, and acceptable propagation lengths, making them the ideal options for a range of ultra-compact devices. Later versions of the MIM device allow for the execution of several logic operations without affecting the phase of the input signals [37–45].

In communication systems, all-optical multiplexer (MUX) devices are often used to transport more signals across a shared medium instead of N channels. Modeling optical shift registers and optical arithmetic logical units (ALUs) necessitates the use of an all-optical MUX [46]. In order to handle the fast-growing data load, many optical multiplexing strategies have been tried during the last few decades. T- and Y-shaped photonic crystals were used in an earlier proposal for a photonic crystal-based MUX; however, the device lacked speed and the wafer size was huge (7.8 mm × 9 mm) [47–49]. When compared to earlier works using different structures, such as heterostructure photonic crystals ring resonators, plasmonic circular ring resonators, and square photonic crystals ring resonators, the proposed model of all-optical 2 × 1 MUX is designed by the plasmonic Y-shaped MIM waveguide based on the principle of linear interference.

The contents of this study are as follows: Section 2 examines the design and operation of an all-optical MUX; Section 3 describes the simulation results; and analyses the performance of the recommended structure; and Section 4 provides a conclusion.

## 2. Design and Operating Principle of 2 × 1 Multiplexer

In fabrication of complex digital systems, a transmit line is required to transport many digital signals; however, only one signal may be delivered at once. In this situation, a tool is required to choose the signals that are likely to be sent on this shared line at various times. A multiplexer is a device with the function of selecting any one of the 'n' inputs and producing a single output. As a consequence, they are sometimes referred to as data selectors. It is a multi-input, single-output switch or device that boosts the data transmission rates that are possible across a shared channel. A MUX may accommodate up to 2N input lines, N select lines, and one output line.

A 2 × 1 MUX has two inputs ($A_0$ and $A_1$), one output (Y), and one control signal input (S). The control signal line selects one of the input lines to send data to the output line. The existence or absence of a light signal at the output (Y) while the control signal (S) is inactive relies on the input signal ($A_0$). The input signal $A_1$ determines whether there is a light signal at output (Y) while the control signal (S) is active. The truth table of 2 × 1 multiplexer is depicted in Table 1.

The schematic of the proposed 2 × 1 multiplexer is depicted in Figure 1 that consists of four linear waveguides each of 2.8 μm and four S-bend waveguides each of 3.6 μm arranged in Y-shape, separated by a distance of 2.6 μm along XZ axis designed on a minimum wafer size of 6 μm × 4 μm. In the structure put forward, the upper linear arm is taken as input $A_0$, the lower linear arm is taken as input $A_1$ and middle linear arm is considered as control signal, S which is sent to both the arms of the designed MUX simultaneously.

The structure is designed on a plasmonic MIM waveguide of continuous-waveform (CW) in transverse electric (TE) mode, with wavelength (λ) of 1.55 μm provided at both inputs. The optical intensity at input for low and high intensity are $0.7 \times 10^9$ W/m and $3 \times 10^9$ W/m, respectively, as tabulated in Table 2 along with various simulation parameters of the proposed 2 × 1 MUX design.

**Table 1.** Truth table of 2 × 1 multiplexer.

| Control Signal (S) | Inputs | | Output |
|---|---|---|---|
| | $A_0$ | $A_1$ | Y |
| 0 | 0 | 0 | 0 |
| | 0 | 1 | 0 |
| | 1 | 0 | 1 |
| | 1 | 1 | 1 |
| 1 | 0 | 0 | 0 |
| | 0 | 1 | 1 |
| | 1 | 0 | 0 |
| | 1 | 1 | 1 |

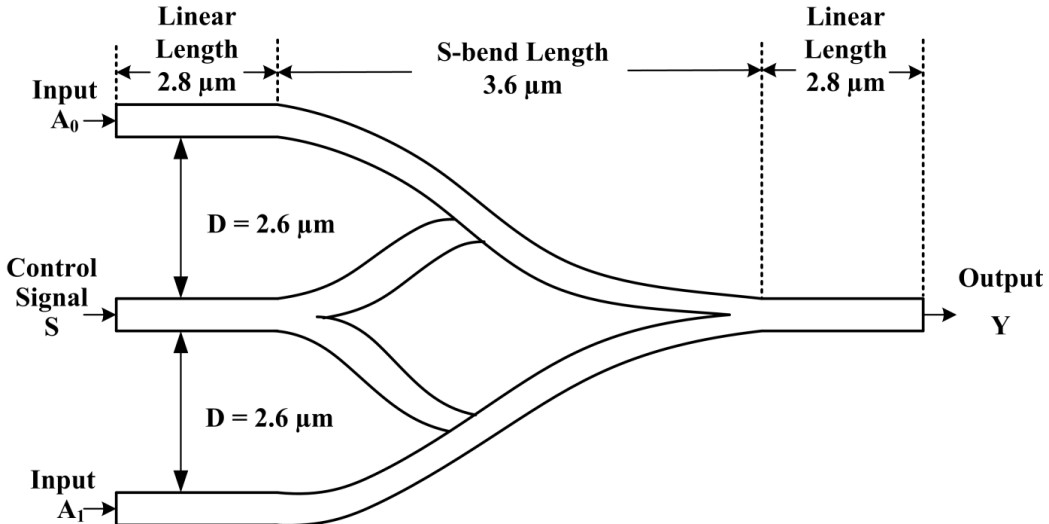

**Figure 1.** Schematic of the proposed 2 × 1 multiplexer.

**Table 2.** Simulation parameters of proposed 2 × 1 multiplexer.

| Simulation Parameters | Considered Value |
|---|---|
| Low power intensity | $0.7 \times 10^9$ W/m |
| High power intensity | $3 \times 10^9$ W/m |
| X mesh cells | 349 |
| Z mesh cells | 603 |
| Transverse Input field | Gaussian |
| Simulation type | 2D |
| Mesh size | 0.0114 μm (X)/0.0114 μm (Y) |
| Boundary conditions | Anisotropic perfectly matched layer (PML) |
| Time Step size | $9.77 \times 10^{17}$ |
| Anisotropic PML layer number | 10 |
| Theoretical reflection coefficient | $1.0 \times 10^{12}$ |
| Real Anisotropic PML tensor parameter | 5 |
| Power of grading polynomial | 3.5 |

By comparing the output power intensity of various refractive indices (n), 2.1 with Boron Nitride material is chosen for the proposed design of 2 × 1 MUX, whereas air is chosen as the dielectric material with an refractive index of 1, as tabulated below in Table 3.

**Table 3.** Output power analysis of 2 × 1 multiplexer with various refractive indices (n).

| 2 × 1 Multiplexer Conditions | Output Power | | |
|:---:|:---:|:---:|:---:|
| | n = 2.05 | n = 2.1 | n = 2.15 |
| 1 | 0.4 | 0.2 | 0.3 |
| 2 | 0.58 | 0.3 | 0.4 |
| 3 | 0.4 | 0.58 | 0.43 |
| 4 | 0.9 | 1.22 | 0.8 |
| 5 | 0.01 | 0.08 | 0.05 |
| 6 | 3.12 | 6.22 | 2.3 |
| 7 | 0.3 | 0.14 | 0.25 |
| 8 | 0.7 | 1.48 | 0.83 |

At a given instance, the selection input is fixed as 0 or 1 and the output is verified for all the input cases by changing the orientation of the phase between the applied inputs and the extinction ratio (ER) and insertion loss (IL) are calculated, respectively. The ER is estimated by comparing the peak output power in $ON$ $\left( P_{out \mid ON} \right)$ with peak output power in OFF $(P_{out \mid OFF})$ states and is represented as

$$\text{ER} = 10 \, log_{10} \left( \frac{P_{out \mid ON}}{P_{out \mid OFF}} \right) \tag{1}$$

whereas the Insertion loss (IL) is defined as the ratio of total input power $(P_{in})$ to the total output power $(P_{out})$, which is given as

$$\text{IL} = 10 \, log_{10}(P_{in} \mid P_{out}) \tag{2}$$

The results are utilised to calculate performance measures, such as IL and ER.

### 3. Simulation and Results of the Proposed 2 × 1 Multiplexer

The Opti-FDTD approach, which takes advantage of device analysis, employs a continuous optical TE wave along the exactly matched circumstances at the input. The simulation results are shown in Figure 2a for a variety of input condition combinations (h). The interference effect governs the phase of light beams under the logical condition "1," which simplifies the design of any logic operation. Phases of the light beams at the various input ports are chosen to enable interference to occur either way. When two optical waves have a phase difference greater than 2n, where n = 0, 1, 2, etc., constructive interference, according to the wave optics theory, develops. The output, which corresponds to the logic state "1," has a high degree of power. Destructive interference occurs when the phase difference is (2n + 1), leading to logic 0 at the output port. In this design, logic "1" is defined as $3 \times 10^9$ W/m and logic "0" as $0.7 \times 10^9$ W/m. All input states and the control signal input get a change in the input phase with either 0° or 180° to fulfill the gate's output. The suggested 2 × 1 Multiplexer's simulation parameters are shown in Table 2.

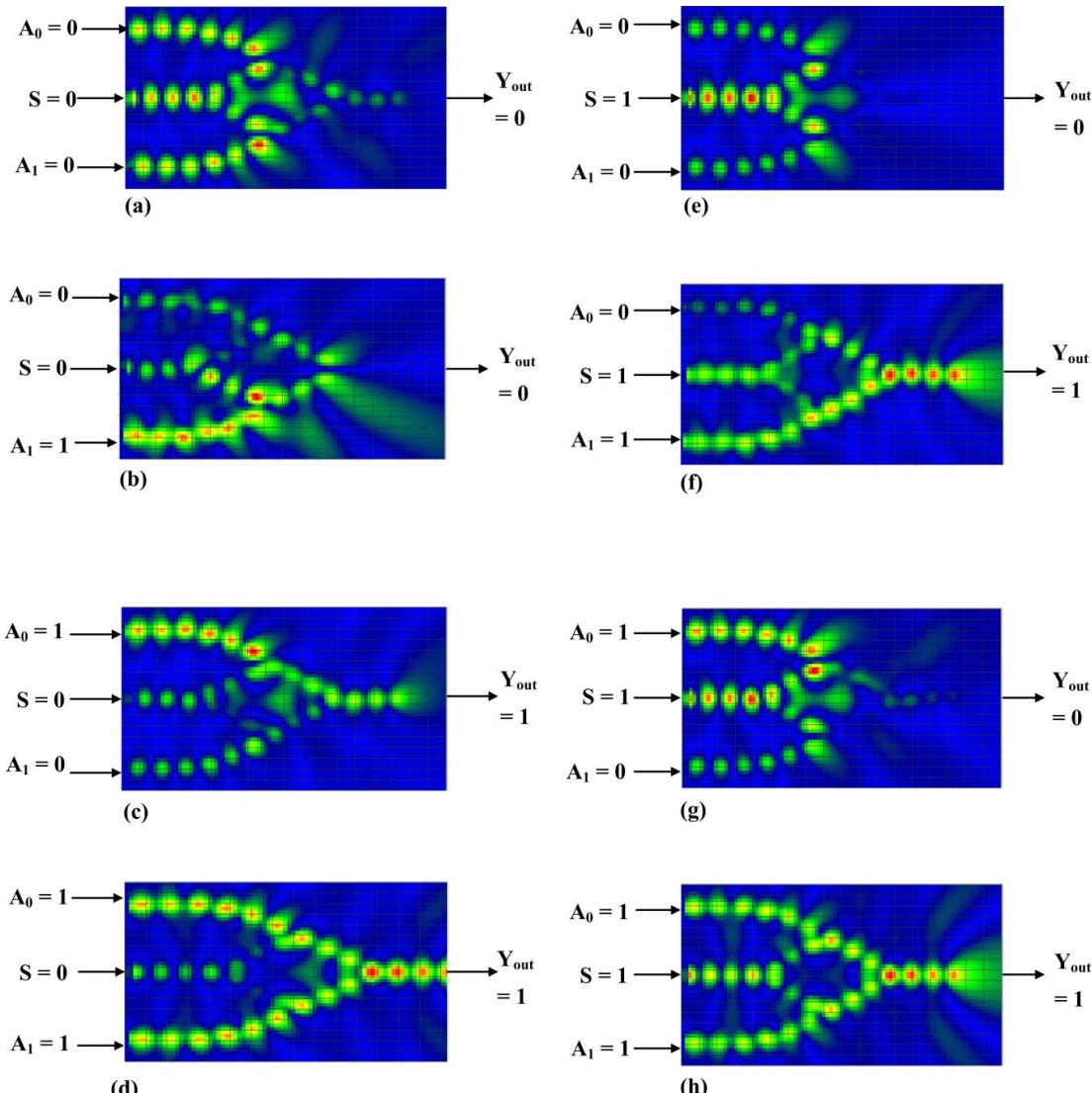

**Figure 2.** (**a**–**h**). Propagation of light through the 2 × 1 multiplexer for various conditions of control signal(S) and inputs $A_0$, $A_1$ using opti-FDTD method.

Condition 1: For control signal, S = 0, $A_0$ = 0, $A_1$= 0.

The existence or absence of a light signal at the output (Y) while the control signal (S) is inactive relies on the input signal ($A_0$). Since the light signal at the output port is similarly zero when $A_0$ = 0, it is noticed that the intensity of light at port Y is exactly correlated with the signal at the input port, as shown in the previous Figure 2a.

Condition 2: For control signal, S = 0, $A_0$ = 0, $A_1$= 1.

When the control signal (S) is inactive, the presence or absence of the light signal at output (Y) depends on the input signal $A_0$. Therefore, when $A_0$ = 0, the light signal at the output port, Y is also 1. Thus, the intensity of light at the port Y is observed to be directly dependent on input port signal $A_0$, as shown in the above Figure 2b.

Condition 3: For control signal, S = 0, $A_0$ = 1, $A_1$= 0.

The existence or absence of a light signal at the output (Y) while the control signal (S) is inactive relies on the input signal ($A_0$). The intensity of light at port Y is discovered to be directly dependent on input port signal $A_0$, as shown in the above Figure 2c. When $A_0$ = 1, the light signal at the output port is likewise 1; therefore, the relationship between the two is clear.

Condition 4: For the control signal, S = 0, $A_0$ = 1, $A_1$= 1.

In the absence of the control signal (S), the input signal $A_0$ determines whether there will be a light signal at the output(Y). The intensity of light at port Y is observed to be directly dependent on the input port signal $A_0$, as illustrated in the above Figure 2d, because when $A_0 = 1$, the light signal at the output port is likewise 1.

Condition 5: For control signal, S = 1, $A_0 = 0$, $A_1 = 0$

The input signal $A_1$ determines whether or not there is a light signal at output (Y) while the control signal (S) is active. The intensity of light at port Y is discovered to be directly dependent on the input port signal $A_1$, as illustrated in the above Figure 2e, because when $A_1 = 0$, the light signal at the output port, Y, is likewise 0.

Condition 6: For the control signal, S = 1, $A_0 = 0$, $A_1 = 1$.

The presence or absence of a light signal at the output (Y) when the control signal (S) is active relies on the input signal $A_1$. The intensity of light at port Y is discovered to be directly dependent on input port signal $A_1$, as illustrated in the above Figure 2f, because when $A_1 = 1$, the light signal at port Y is also 1.

Condition 7: For control signal, S = 1, $A_0 = 1$, $A_1 = 0$.

When the control signal (S) is active, the presence or absence of the light signal at output (Y) depends on the input signal $A_1$. When $A_1 = 1$, the light signal at the output port is Y = 1; therefore, the intensity of light at the port Y is observed to be directly dependent on the input port signal $A_1$, as shown in the above Figure 2g.

Condition 8: For the control signal, S = 1, $A_0 = 1$, $A_1 = 1$.

The input signal $A_1$ determines whether there is a light signal at output (Y) while the control signal (S) is active. As can be shown in Figure 2h, the intensity of light at port Y is observed to be directly dependent on the signal from the input port when $A_1 = 1$, and the light signal at the output port is also 1 when $A_1 = 1$.

To analyze the intensity of the light propagated, Figure 3 depicting the color bar, indicating the intensity of light, is given below.

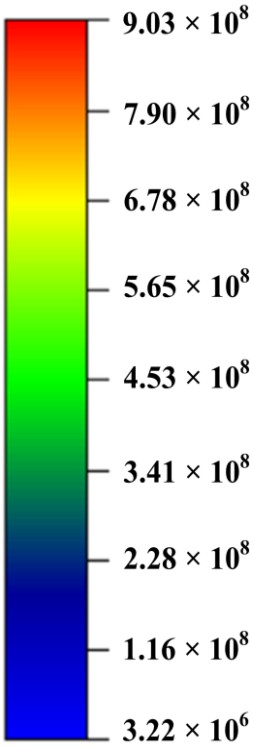

**Figure 3.** Color bar indicating the intensity of light.

Table 4 indicates the values of output power ($P_Y$) obtained by providing a optical input power signal at a low intensity of $0.7 \times 10^9$ W/m and high intensity of $3 \times 10^9$ W/m for the inputs ($A_0$ and $A_1$) and control signal (S). A normalized power of 0.5 is used as

the threshold for the proposed $2 \times 1$ MUX, below which the output is regarded as a low-intensity signal and beyond which it is regarded as a high-intensity signal, which are simulated in 2D using the opti-FDTD software.

**Table 4.** Truth table of Multiplexer in terms of logic power output values with input power at low intensity of $0.7 \times 10^9$ W/m and high intensity of $3 \times 10^9$ W/m.

| Control Signal (S) | Inputs | | Output | Logic Power Output |
|---|---|---|---|---|
| | $A_0$ | $A_1$ | Y | $P_Y$ |
| 0 | 0 | 0 | 0 | 0.20 |
| | 0 | 1 | 0 | 0.40 |
| | 1 | 0 | 1 | 0.58 |
| | 1 | 1 | 1 | 1.22 |
| 1 | 0 | 0 | 0 | 0.08 |
| | 0 | 1 | 1 | 6.22 |
| | 1 | 0 | 0 | 0.14 |
| | 1 | 1 | 1 | 1.48 |

## 4. Conclusions

In conclusion, utilising 2D FDTD approaches, a new $2 \times 1$ multiplexer was designed and studied, which is built on a Y-shaped plasmonic MIM structure. Plasmonic devices, which enable nanophotonics and nanodevices, offer a solution to the miniaturisation and diffraction limit problems in photonics devices. The proposed device works by using interference between input signals and selector signals, which may be both destructive and beneficial. Performance of the proposed device is evaluated using elements such as transmission and extension ratio.

**Author Contributions:** Conceptualization, S.S. and I.C.; methodology, V.P.; software, G.R.N.; validation, S.S. and S.K.; formal analysis, I.C., S.S. and V.P.; writing—original draft preparation, I.C., S.S., G.R.N. and V.P.; writing—review and editing, S.S. and S.K.; supervision, S.S. and S.K.; All authors have read and agreed to the published version of the manuscript.

**Funding:** This research received no external funding.

**Institutional Review Board Statement:** Not Applicable.

**Informed Consent Statement:** Not Applicable.

**Data Availability Statement:** Not Applicable.

**Acknowledgments:** This work was supported by the Double–Hundred Talent Plan of Shandong Province, China.

**Conflicts of Interest:** The authors declare no conflict of interest.

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
