# Peer review of "An All Optical 2 × 1 Multiplexer Using a Metal-Insulator-Metal based Plasmonic Waveguide for Processing at a Rapid Pace"

_photonics, doi:10.3390/photonics10010074_

Round 1

Reviewer 1 Report

In the present manuscript, the authors propose a 2×1 multiplexer (MUX) in simulation, which is based on two plasmonic Y-shaped MIM waveguides arranged facing each other. The authors tuned the design parameters to optimize performance of the plasmonic multiplexer. Based on the calculated results, the authors gave some detail analysis of the performance characteristics of the design MUX, such as extinction ratio and insertion loss. However, the manuscript is not well organized and the results are insufficient practice. I think the manuscript should give major revision before reconsideration in Photonics. I have some detail comments as follow.

(1) Sections 2 and 3 are not well organized. In this version, the manuscript looks like a research note, instead of a research article.

(2) Which type of metal are chosen in this work? The effects on the loss of metal should be discussed, particular for the performance of MUX.

(3) In figure 2, the colorbar should be given, such that the readers can determine the light intensity.

(4) As a simulation word, to ensure the readers can repeat the design MUX, the authors should give more details on the design and simulations, such as the setup of boundary condition and detection in simulated model, the curvature radius and arc angle of S-bend waveguides.

(5) Why did choose the refractive index of the core of the plasmonic waveguides as 2.1? Is such value important for the performance of MUX? Which material can have the refractive index of 2.1 at telecommunication wavelength?

(6) How about the transmission spectrum of the MUX device? When the geometry parameters are deviated from the design value, what will the performance of MUX change?

Author Response

Date: Dec. 18, 2022

To,

Editor,

Photonics

Subject: Submission of MAJOR REVISION of Manuscript - photonics-2050800

Dear Sir,                                            

We express our sincere thanks to your editorial team comprising of the editor, associate editor, and reviewers for providing us with the opportunity of revising the manuscript.

We have carefully analyzed all the technical comments and have addressed all of them with the utmost care. The modified sections in the manuscript are highlighted in Red-colored font. The specific replies to the queries of the reviewers are also addressed below.

We sincerely believe that these changes have indeed improved the quality of our present submission and satisfy the reviewers and the editorial team of your esteemed journal.

Thanking you,

Yours sincerely,

Dr. Sandip Swarnakar

(On behalf of all the authors)

Manuscript ID: photonics-2050800

Title: An all optical 2×1 multiplexer using a Metal-Insulator-Metal based plasmonic waveguide for processing at a rapid pace

 Reply to the Specific Comments of Reviewer's

Reviewer: 1

In the present manuscript, the authors propose a 2×1 multiplexer (MUX) in simulation, which is based on two plasmonic Y-shaped MIM waveguides arranged facing each other. The authors tuned the design parameters to optimize performance of the plasmonic multiplexer. Based on the calculated results, the authors gave some detail analysis of the performance characteristics of the design MUX, such as extinction ratio and insertion loss. However, the manuscript is not well organized and the results are insufficient practice. I think the manuscript should give major revision before reconsideration in Photonics. I have some detail comments as follow.
Response:

We are greatly thankful to you for your kind words of appreciation. Your queries indeed have helped us to improve the quality of the manuscript. We have further clarified your comment, as mentioned below.

Comment 1: Sections 2 and 3 are not well organized. In this version, the manuscript looks like a research note, instead of a research article.

Response: We are thankful to reviewer for your valuable suggestion; it will improve the manuscript’s clarity. According to your suggestion manuscript has been modified in the sections 2 and 3 is indicated by using red-color.

Comment 2: Which type of metal is chosen in this work? The effects on the loss of metal should be discussed, particular for the performance of MUX.

Response: Thank you very much for your valuable suggestion; it will improve the manuscript’s clarity. The metal material used in this simulation is Boron nitride with refractive index 2.1. The manuscript is modified according to your suggestions and is copied below for your ready reference (section 2). A copy of the updated manuscript is given below for ready reference.

By comparing the output power intensity of various refractive indices (n), 2.1 with Boron Nitride material is chosen for the proposed design of 2x1 MUX whereas air is chosen as dielectric material with an refractive index of 1 as tabulated below in table 3.

Comment 3: In figure 2, the color bar should be given, such that the readers can determine the light intensity.

Response:

Thank you very much for this valuable comment. As suggested by the reviewer, we have added the color bar in the figure 3 of section 3. A copy of the updated manuscript is given below for ready reference.

To analyze the intensity of light propagated, figure 3 depicting color bar indicating the intensity of light is given below.

Figure 3. Color bar indicating the intensity of light.

Comment 4: As a simulation word, to ensure the readers can repeat the design MUX, the authors should give more details on the design and simulations, such as the setup of boundary condition and detection in simulated model, the curvature radius and arc angle of S-bend waveguides.

Response:

Thank you very much for this valuable comment. In this manuscript as per your suggestion we have added more details on the design and simulation. A copy of the updated manuscript is shown below.  

Table 2.Simulation parameters of proposed 2×1 multiplexer.

Simulation Parameters

Considered value

Less power intensity

0.7 × 109 W/m

High power intensity

3 × 109 W/m

X mesh cells

349

Z mesh cells

603

Transverse Input field

Gaussian

Simulation type

2D

Mesh size

0.0114 µm (X)/ 0.0114 µm (Z)

Boundary conditions

Anisotropic perfectly matched layer (PML)

Time Step size

9.77e-17

Anisotropic PML layer number

10

Theoretical reflection coefficient

1.0e-12

Real Anisotropic PML tensor parameter

5

Power of grading polynomial

3.5

In order to divert the optical signal in the required direction, the normal waveguide has been bent in the shape of “S” where the radius of the curvature or the bent angle should not be less than the twice of the wavelength being used. The wavelength being used for this simulation is 1550 nm, so the radius of curvature of the S-bend waveguide should be around 3100 nm. The angle between the S-bend waveguides has been calculated by drawing a tangent to the since wave at the point where the waveguide crosses the X-axis. It is found that the waveguide intersects the x-axis at P=4.063 and tangent makes an inclination angle of 140 with the positive axis. Hence the total angle between the S-bend equals to 280. The same has been depicted in the below fig

Comment 5: Why did choose the refractive index of the core of the plasmonic waveguides as 2.1? Is such value important for the performance of MUX? Which material can have the refractive index of 2.1 at telecommunication wavelength?

Response:

Thank you very much for this valuable comment. The metal material used in this simulation is Boron nitride with refractive index 2.1 with a wavelength of 1.55 µm as it is an excellent host for confined light and suppression of plasmon losses. The manuscript is modified according to your suggestions and is copied below for your ready reference (section 2). A copy of the updated manuscript is given below for ready reference.

The structure is designed on a plasmonic waveguide of refractive index (n) as 2.1 with Boron Nitride material and continuous-waveform (CW) in transverse electric (TE) mode, with wavelength (λ) of 1.55 µm provided at both inputs.

Comment 6: How about the transmission spectrum of the MUX device? When the geometry parameters are deviated from the design value, what will the performance of MUX change?

Response:

We are thankful to reviewer for valuable suggestions, and we understood that suggestions help us to improve the quality of the manuscript. As mentioned in table 2 when the parameters are deviated the desired output is not obtained resulting in huge dispersion in light.

Reviewer 2 Report

An all optical 2×1 multiplexer using a Metal-Insulator-Metal based plasmonic waveguide for processing at a rapid pace", by I. Charles et al., reported a novel plasmonic Y-shaped MIM waveguide, which demonstrate superior performace. The experiment is wll designed and the conclusions are well supported. For that, I'd like to recommend for publication as it is.

Recommendation: Accept in present form

Author Response

Date: Dec. 18, 2022

To,

Editor,

Photonics

Subject: Submission of MAJOR REVISION of Manuscript - photonics-2050800

Dear Sir,                                            

We express our sincere thanks to your editorial team comprising of the editor, associate editor, and reviewers for providing us with the opportunity of revising the manuscript.

We have carefully analyzed all the technical comments and have addressed all of them with the utmost care. The modified sections in the manuscript are highlighted in Red-colored font. The specific replies to the queries of the reviewers are also addressed below.

We sincerely believe that these changes have indeed improved the quality of our present submission and satisfy the reviewers and the editorial team of your esteemed journal.

Thanking you,

Yours sincerely,

Dr. Sandip Swarnakar

(On behalf of all the authors)

Manuscript ID: photonics-2050800

Title: An all optical 2×1 multiplexer using a Metal-Insulator-Metal based plasmonic waveguide for processing at a rapid pace

 Reply to the Specific Comments of Reviewer's

Reviewer: 2

An all-optical 2×1 multiplexer using a Metal-Insulator-Metal based plasmonic waveguide for processing at a rapid pace", by I. Charles et al., reported a novel plasmonic Y-shaped MIM waveguide, which demonstrate superior performace. The experiment is well designed and the conclusions are well supported. For that, I'd like to recommend for publication as it is.

We are greatly thankful to you for your kind words of appreciation and recommending this manuscript to be accepted for publication.

Reviewer 3 Report

The authors proposed 2×1 multiplexer using a Metal-Insulator-Metal waveguide. Although this manuscript is interesting, there are many problems, such as English problems, that need to be carefully addressed:

¾   The abstract is not written scientifically and correctly.

¾   The conclusion is not written scientifically and correctly.

¾   There are inappropriate self-citations by the authors.

¾   The English is poor in this manuscript; there are many English problems and typing errors.

¾   The fabrication process of the structure should explain by the authors.

¾   What is the material of the waveguide in Fig. 1?

¾   What is the material of the substrate in Fig. 1?

¾   Why 0.7e9 W/m is considered as logic “0”?

¾   In Fig. 2, the authors should mention the exact value of outport power (W/m).

¾   There is no explanation in the manuscript for Tables 1, 3, and 4.

¾   The authors said the outport power for logic “1” is 3e9 W/m. But in Table 4, the output powers of 0.58, 1.22, and 1.48 are considered as logic “1”!

¾   The simulation is 2D or 3D? can authors prove their results with 3D simulations?

¾   In regards to the optical multiplexers and demultiplexers, the authors should mention other configurations in the introduction section such as:

        o "Design of photonic crystal based compact all-optical 2× 1
multiplexer for optical processing devices." Microelectronics Journal
112 (2021): 105046.           o "An ultra-fast all-optical 2-to-1 digital multiplexer based on photonic crystal ring resonators." Optical and Quantum Electronics 54.7 (2022): 1-12.           o "Design and simulation of wavelength demultiplexer based on
heterostructure photonic crystals ring resonators." Physica E:
Low-dimensional Systems and Nanostructures 50 (2013): 97-101.

        o "Quasi-phase-matching-division multiplexing holography in a three-dimensional nonlinear photonic crystal." Light: Science & Applications 10.1 (2021): 1-7.  

        o "Heterostructure four channel wavelength demultiplexer using square
photonic crystals ring resonators." Journal of Electromagnetic waves and
Applications 26.13 (2012): 1700-1707.

Author Response

Date: Dec. 18, 2022

To,

Editor,

Photonics

Subject: Submission of MAJOR REVISION of Manuscript - photonics-2050800

Dear Sir,                                            

We express our sincere thanks to your editorial team comprising of the editor, associate editor, and reviewers for providing us with the opportunity of revising the manuscript.

We have carefully analyzed all the technical comments and have addressed all of them with the utmost care. The modified sections in the manuscript are highlighted in Red-colored font. The specific replies to the queries of the reviewers are also addressed below.

We sincerely believe that these changes have indeed improved the quality of our present submission and satisfy the reviewers and the editorial team of your esteemed journal.

Thanking you,

Yours sincerely,

Dr. Sandip Swarnakar

(On behalf of all the authors)

Manuscript ID: photonics-2050800

Title: An all optical 2×1 multiplexer using a Metal-Insulator-Metal based plasmonic waveguide for processing at a rapid pace

 Reply to the Specific Comments of Reviewer's

Reviewer: 3

The authors proposed 2×1 multiplexer using a Metal-Insulator-Metal waveguide. Although this manuscript is interesting, there are many problems, such as English problems, that need to be carefully addressed:

Response: Thank you very much for your suggestion. According to your suggestion manuscript has been modified and it is indicated by using red-color.

Comment 1: The abstract is not written scientifically and correctly.

Response: Thank you very much for your suggestion. Based on your suggestion, abstract is modified likewise in the manuscript.

This study proposes, designs, and simulates a unique plasmonic Y-shaped MIM waveguide based 2x1 multiplexer (MUX) structure utilising opti-FDTD software. Two plasmonic Y-shaped waveguides are positioned facing one another inside a minimum wafer size of 6 µm × 3.5 µm in the 2x1 MUX configurations that is being described. The design parameters are adjusted until the plasmonic multiplexer performs as required under optimal conditions. Extinction ratio and insertion loss are two performance metrics that are calculated for performance analysis of the design which indicate the potential to be applied in plasmonic integrated circuits.

Comment 2: The conclusion is not written scientifically and correctly.

Response: We are thankful to reviewer for valuable suggestions, and we understood that suggestions help us to improve the quality of the manuscript. We have modified the conclusion part of the manuscript as per reviewer’s suggestions.

In conclusion, utilising 2D opti FDTD approaches, a new 2x1 multiplexer was designed and studied which is built on a Y-shaped plasmonic MIM structure. Plasmonic devices, which enable nanophotonics and nanodevices, offer a solution to the miniaturization and diffraction limit problems in photonics devices. The proposed device works by using interference between input signals and selector signals, which may be both destructive and beneficial. Performance of the proposed device is evaluated using elements such as transmission and extension ratio.

Comment 3: There are inappropriate self-citations by the authors.

Response: Thank you very much for your suggestion. According to your suggestion, the manuscript has been modified.

Comment 4: The English is poor in this manuscript; there are many English problems and typing errors.

Response: Thank you very much for your suggestion. According to your suggestion the manuscript is verified and modified.

Comment 5: The fabrication process of the structure should explain by the authors.

Response: Thank you very much for these comments. As per your suggestions, the fabrication process is mentioned in the modified manuscript.  The modified manuscript is copied below for ready reference.

The plasmonic MIM waveguide which we have used in this proposed design is a combination of Air as the dielectric material sandwiched between Boron Nitride (crystalline-BN) as the metallic layers. These multi-layered based waveguide components are practically realizable as they are fully compatible with the existing semiconductor fabrication magnetron sputtering and Plasma Enhanced Chemical Vapor Disposition (PECVD). The present work entitled “An all optical 2×1 multiplexer using a Metal-Insulator-Metal based plasmonic waveguide for processing at a rapid pace” is done by considering the 2D parameters since our proposed work is a 2-dimensional design.

Comment 6: What is the material of the waveguide in Fig. 1?

Response: Thank you very much for this valuable comment. The metal material used in this simulation is Boron nitride with refractive index 2.1. The manuscript is modified according to your suggestions and is copied below for your ready reference (section 2). A copy of the updated manuscript is given below for ready reference.

The structure is designed on a plasmonic waveguide of refractive index (n) as 2.1 with Boron Nitride material and continuous-waveform (CW) in transverse electric (TE) mode, with wavelength (λ) of 1.55 µm provided at both inputs.

 Comment 7: What is the material of the substrate in Fig. 1?

Response: Thank you very much for this valuable comment. The metal material used in this simulation is Boron nitride with refractive index 2.1. The manuscript is modified according to your suggestions and is copied below for your ready reference (section 2). A copy of the updated manuscript is given below for ready reference.

The structure is designed on a plasmonic waveguide of refractive index (n) as 2.1 with Boron Nitride material and continuous-waveform (CW) in transverse electric (TE) mode, with wavelength (λ) of 1.55 µm provided at both inputs.

Comment 8: Why 0.7e9 W/m is considered as logic “0”?

Response: Thank you for your comment. According to Ref [26], the signal strength for logic 0 and logic 1 are considered as 0.7e9 W/m and 3e9 W/m respectively, due to the best results observed from the literature survey. The manuscript is copied below for your ready reference.

[29] Fakhruldeen, H.F.; Mansour, T.S. Design of Plasmonic NOT Logic Gate Based on Insulator–Metal–Insulator (IMI) waveguides. Advanced Electromagnetics 2020, 9(1),91-94. https://doi.org/10.7716/aem.v9i1.1376.

Comment 9: In Fig. 2, the authors should mention the exact value of outport power (W/m).

Response: Thank you very much for these comments. As per your suggestions, the exact values of output power (W/m) are tabulated and mentioned in the modified manuscript.  The modified manuscript is copied below for ready reference.

Table 4 indicates the values of output power (PY) obtained by providing optical input power signal at low intensity of 0.7 × 109 W/m and high intensity of 3 × 109 W/m for the inputs ( A0 and A1 ) and control signal (S). A normalised power of 0.5 is used as the threshold for the proposed 2x1 MUX, below which the output is regarded as a low-intensity signal and beyond which it is regarded as a high-intensity signal are simulated in 2D using the opti-FDTD software.

Table 4. Truth table of Multiplexer in terms of logic power output values with input power at low intensity of 0.7 × 109 W/m and high intensity of 3 × 109 W/m.

Control signal (S)

Inputs

Output

Logic power output

A0

A1

Y

PY

0

0

0

0

0.20

0

1

0

0.40

1

0

1

0.58

1

1

1

1.22

1

0

0

0

0.08

0

1

1

6.22

1

0

0

0.14

1

1

1

1.48

Comment 10: There is no explanation in the manuscript for Tables 1, 3, and 4.

Response: Thank you very much for these comments. As per your suggestions, the explanation for the tables 1, 3 and 4 are added in the modified manuscript.  The modified manuscript is copied below for ready reference.

A 2×1 MUX has two inputs (A0 and A1), one output (Y), and one control signal input (S). The control signal line selects one of the input lines to send data to the output line. The existence or absence of a light signal at the output (Y) while the control signal (S) is inactive relies on the input signal (A0). The input signal A1 determines whether there is a light signal at output (Y) while the control signal (S) is active. The truth table of 2x1 multiplexer is depicted in table 1.

The structure is designed on a plasmonic MIM waveguide of continuous-waveform (CW) in transverse electric (TE) mode, with wavelength (λ) of 1.55 µm provided at both inputs. The optical intensity at input for low and high intensity are 0.7 × 109 W/m and 3×109W/m, respectively as tabulated in table 2 along with various simulation parameters of proposed 2x1 MUX design.

            Table 2.Simulation parameters of proposed 2×1 multiplexer.

Simulation Parameters

Considered value

Low power intensity

0.7 × 109 W/m

High power intensity

3 × 109 W/m

X mesh cells

349

Z mesh cells

603

Transverse Input field

Gaussian

Simulation type

2D

Mesh size

0.0114 µm (X)/ 0.0114 µm (Y)

Boundary conditions

Anisotropic perfectly matched layer (PML)

Time Step size

9.77e-17

Anisotropic PML layer number

10

Theoretical reflection coefficient

1.0e-12

Real Anisotropic PML tensor parameter

5

Power of grading polynomial

3.5

By comparing the output power intensity of various refractive indices (n), 2.1 with Boron Nitride material is chosen for the proposed design of 2x1 MUX whereas air is chosen as dielectric material with an refractive index of 1 as tabulated below in table 3.

Comment 11: The authors said the outport power for logic “1” is 3e9 W/m. But in Table 4, the output powers of 0.58, 1.22, and 1.48 are considered as logic “1”!

Response: Thank you very much for these comments. In the suggested design, for logic high(1) 3e9 W/m and for logic low(0) 0.7e9 W/m are taken at input port as input power intensity and the power(PY) is observed at output port by considering a normalized power of 0.5 and above as logic high. As per your suggestions, detailed modified manuscript is copied below for ready reference.

Table 4 indicates the values of output power (PY) obtained by providing optical input power signal at low intensity of 0.7 × 109 W/m and high intensity of 3 × 109 W/m for the inputs ( A0 and A1 ) and control signal (S). A normalised power of 0.5 is used as the threshold for the proposed 2x1 MUX, below which the output is regarded as a low-intensity signal and beyond which it is regarded as a high-intensity signal are simulated in 2D using the Opti-FDTD software.

Comment 12: The simulation is 2D or 3D? Can authors prove their results with 3D simulations?

Response: Thank you very much for these comments. As mentioned in table 2, the simulation for the proposed design is done in 2D and the table is given below for your reference with some added simulation details. At present 3D simulation is not planned to be done for this proposed work.

Table 2.Simulation parameters of proposed 2×1 multiplexer.

Simulation Parameters

Considered value

Low power intensity

0.7 × 109 W/m

High power intensity

3 × 109 W/m

X mesh cells

349

Z mesh cells

603

Transverse Input field

Gaussian

Simulation type

2D

Mesh size

0.0114 µm (X)/ 0.0114 µm (Y)

Boundary conditions

Anisotropic perfectly matched layer (PML)

Time Step size

9.77e-17

Anisotropic PML layer number

10

Theoretical reflection coefficient

1.0e-12

Real Anisotropic PML tensor parameter

5

Power of grading polynomial

3.5

Comment 13: In regards to the optical multiplexers and demultiplexers, the authors should mention other configurations in the introduction section such as:

  1. "Design of photonic crystal based compact all-optical 2× 1 multiplexer for optical processing devices." Microelectronics Journal 112 (2021): 105046.
  2. "Design and simulation of wavelength demultiplexer based on heterostructure photonic crystals ring resonators." Physica E: Low-dimensional Systems and Nanostructures 50 (2013): 97-101.
  3. "Design and optimization of all-optical demultiplexer using photonic crystals for optical computing applications." Journal of Optical Communications (2020).
  4. "Dual wavelength demultiplexer based on metal–insulator–metal plasmonic circular ring resonators." Journal of Modern Optics 63.11 (2016): 1078-1086.
  5. "Heterostructure four channel wavelength demultiplexer using square photonic crystals ring resonators." Journal of Electromagnetic waves and Applications26.13 (2012): 1700-1707.

Response:

Thank you very much for this valuable comment. As suggested by the reviewer, we have mentioned other configurations in the introduction section and added additional references in the reference section on the suggested topics. A copy of the updated manuscript is given below for ready reference.

  1. Rao, D.G.S.; Swarnakar, S.; Kumar, S. Design of photonic crystal based compact all-optical 2× 1 multiplexer for optical processing devices. Microelectron. J. 2021, 112, 1-6. https://doi.org/10.1016/j.mejo.2021.105046.
  2. Rakhshani, M.R.; Mansouri-Birjandi, M.A. Design and simulation of wavelength demultiplexer based on heterostructure photonic crystals ring resonators. Physica E: Low-dimensional Systems and Nanostructures 2013, 1(50), 97-101. https://doi.org/10.1016/j.physe.2013.03.003.
  3. Rakhshani, M.R.; Mansouri-Birjandi, M.A. Dual wavelength demultiplexer based on metal–insulator–metal plasmonic circular ring resonators. J. Modern Optics 2016, 63(11), 1078-1086. https://doi.org/10.1080/09500340.2015.1125962.
  4. Rakhshani, M.R.; Mansouri-Birjandi, M.A. Heterostructure four channel wavelength demultiplexer using square photonic crystals ring resonators.  Electromagnetic waves and Applications, 2012 26(13), 1700-1707. https://doi.org/10.1080/09205071.2012.709927.
